# Enhanced Efficiency of the Basal and Induced Apoptosis Process in Mucopolysaccharidosis IVA and IVB Human Fibroblasts

**DOI:** 10.3390/ijms241814119

**Published:** 2023-09-14

**Authors:** Joanna Brokowska, Lidia Gaffke, Karolina Pierzynowska, Grzegorz Węgrzyn

**Affiliations:** Department of Molecular Biology, Faculty of Biology, University of Gdansk, Wita Stwosza 59, 80-308 Gdansk, Poland; joanna.brokowska@phdstud.ug.edu.pl (J.B.); lidia.gaffke@ug.edu.pl (L.G.)

**Keywords:** Morquio disease, mucopolysaccharidosis IV, apoptosis

## Abstract

Morquio disease, also called mucopolysaccharidosis IV (MPS IV), belongs to the group of lysosomal storage diseases (LSD). Due to deficiencies in the activities of galactose-6-sulfate sulfatase (in type A) or β-galactosidase (in type B), arising from mutations in *GALNS* or *GLB1*, respectively, keratan sulfate (one of glycosaminoglycans, GAGs) cannot be degraded efficiently and accumulates in lysosomes. This primary defect leads to many cellular dysfunctions which then cause specific disease symptoms. Recent works have indicated that different secondary effects of GAG accumulation might significantly contribute to the pathomechanisms of MPS. Apoptosis is among the cellular processes that were discovered to be affected in MPS cells on the basis of transcriptomic studies and some cell biology experiments. However, Morquio disease is the MPS type which is the least studied in light of apoptosis dysregulation, while RNA-seq analyses suggested considerable changes in the expression of genes involved in apoptosis in MPS IVA and IVB fibroblasts. Here we demonstrate that cytochrome *c* release from mitochondria is more efficient in MPS IVA and IVB fibroblasts relative to control cells, both under the standard cultivation conditions and after treatment with staurosporine, an apoptosis inducer. This indication of apoptosis stimulation was corroborated by measurements of the levels of caspases 9, 3, 6, and 7, as well as PARP, cleaved at specific sites, in Morquio disease and control fibroblasts. The more detailed analyses of the transcriptomic data revealed which genes related to apoptosis are down- and up-regulated in MPS IVA and IVB fibroblasts. We conclude that apoptosis is stimulated in Morquio disease under both standard cell culture conditions and after induction with staurosporine which may contribute to the pathomechanism of this disorder. Dysregulation of apoptosis in other MPS types is discussed.

## 1. Introduction

Lysosomal storage diseases (LSDs) are a group of several dozens of inherited metabolic disorders, characterized by dysfunctions of lysosomal enzymes and the resultant accumulation of variety of undegraded macromolecules [1]. Mucopolysaccharidoses (MPS) are the largest group of LSDs, consisting of heritable diseases in which lysosomal storage of undegraded glycosaminoglycans (GAGs) is the primary biochemical defect, arising from the genetically determined deficiency in activity of one of enzymes involved in catabolism of these complex carbohydrates [2,3]. Depending on the kind of accumulated GAG(s) and on the nature of deficient enzyme, there are 14 types of MPS described to date [4]. They include types I, II, IIIA, IIIB, IIIC, IIID, IIIE (recognized recently as a deficiency in arylsulfatase G, encoded by the *ARSG* gene [5,6]), IVA, IVB, VI, VII, IX, X (the latest types described [7]), and MPSPS (MPS-Plus syndrome, caused by mutations in the *VPS33A* gene rather than in a gene coding for a lysosomal hydrolase, but resulting in massive GAG accumulation [8,9]).

All MPS types are severe diseases, resulting in dysfunctions in most tissues and organs, though the specific symptoms differ significantly from type to type [2,3,4]. Although it was initially suspected that the major and only biochemical cause of the disease was lysosomal GAG accumulation, subsequent studies clearly indicate that secondary and tertiary cellular changes may be at least as important for the molecular pathomechanism of MPS as the primary storage [10,11,12]. Importantly, apart from secondary biochemical changes and dysmorphology in cellular organelles [13,14], crucial processes occurring in MPS cells were demonstrated to be significantly affected [15]. They include proteasome activity [16], ion homeostasis [17], signal transduction [18], cell cycle [19], vesicle trafficking [20], actin cytoskeleton polymerization [21], and apoptosis [22]. Dysregulation of the last of the mentioned processes has been demonstrated mostly by finding changes in the expression of the genes involved, directly or indirectly, in apoptosis [22]. In addition, increased activity of caspase 3/7 has been found in fibroblasts of some MPS types [22]. These analyses led to the proposal that the apoptosis process may be stimulated in MPS cells, which could contribute to the molecular mechanism of the disease. Such a proposal is also corroborated by results of previous studies in which activation of this programmed cell death in MPS was signaled (as mentioned previously [22] and discussed more deeply in Chapter 3 of this paper).

Intriguingly, despite Morquio disease types A and B being among the MPS types in which the most pronounced changes in the expression of genes related to apoptosis has been found [22], they are the least studied MPS types in light of apoptosis dysregulation (see discussion included in Chapter 3 of this paper). Therefore, the aim of this work was to investigate the apoptosis process in MPS IVA and IVB. Morquio disease is a specific kind of MPS where keratan sulfate (KS) is accumulated as the major stored GAG [23]. Contrary to other MPS types, no (or little) cognitive impairment is observed in the affected patients, while severe skeletal deformities and connective tissue dysfunctions are major symptoms, causing significant disability and considerably shortened (to about 2–3 decades) life span [23]. MPS IVA is caused by mutations in the *GALNS* gene, resulting in a lack of or a significant reduction in the activity of galactose-6-sulfate sulfatase [24]. On the other hand, specific mutation in the *GLB1* gene results in a deficiency in β-galactosidase activity, causing MPS IVB (interestingly, other mutations in this case lead to G_M1_ gangliosidosis) [25]. Both MPS IVA and IVB are inherited in an autosomal recessive manner, and in both of them, KS accumulation is the primary biochemical defect. This leads to the appearance of similar symptoms in patients suffering from both these disorders, though it is commonly described that MPS IVA is a more severe disease than MPS IVB [23]. In this study, we used fibroblasts derived from MPS IVA and IVB patients as cellular models of the diseases. Such cells were previously employed during MPS investigations, indicating that experiments with fibroblasts can give reliable results in studies on molecular mechanisms of MPS [13,14,15,16,17,18,19,20,21,22].

## 2. Results

Previous studies on apoptosis in MPS involved the analysis of transcriptomes of fibroblasts derived from patients with 11 MPS types [22]. It was demonstrated that there are many genes related to this process which are dysregulated in MPS. Among them, MPS IVA and MPS IVB were types with the most pronounced changes in the levels of transcripts of these genes. Here, we analyzed what specific genes’ products, which are involved directly or indirectly, were down- and up-regulated in Morquio disease cells. The levels of transcripts were compared to those occurring in control fibroblasts (the HDFa cell line). This analysis confirmed that there are many apoptosis-related genes in which expression is dysregulated in MPS IVA and IVB cells, including some playing crucial roles in this process, like *CYCS* (coding for cytochrome *c*), *PDK1* (coding for pyruvate dehydrogenase kinase 1), and *XIAP* (coding for X-linked inhibitor of apoptosis) (Table 1). 

The detailed transcriptomic analyses shown in Table 1 confirmed that the regulation of apoptosis may be impaired in MPS IVA and IVB fibroblasts. To test if specific indicators of this process are present in these cells, we investigated the release of cytochrome *c* from mitochondria, the process required to induce apoptosis [26]. A specific anti-cytochrome *c* antibody was used, and mitochondria were visualized by MitoTracker. Cells were cultured without specific treatment or in the presence of staurosporine, an inducer of apoptosis [27], and added to a final concentration of 1 μM for 6 h. In untreated cells, an enhanced release of cytochrome *c* in MPS IVA and IVB cells was observed, relative to control fibroblasts (HDFa) (Figure 1). After the treatment with staurosporine, the features of apoptotic cells were observed in all cell lines; however, the release of cytochrome *c* was again more pronounced in MPS IVA and IVB than in HDFa (Figure 1). These results indicate that apoptosis can be stimulated in Morquio disease, even without specific induction of this process, which corroborated previous suggestions. Moreover, in the presence of staurosporine, apoptosis was still more effective in MPS IVA and IVB cells. 

To learn more about the mechanism of apoptosis stimulation in Morquio disease, we estimated the levels of caspases, cysteine proteases playing crucial roles in this process [28]. Caspases are cleaved at specific sites to be activated, and among several types of these proteases there are those involved in the initiation (caspase-8 and caspase-9) and the execution (caspase-3, caspase-6, and caspase-7) of apoptosis [29,30]. There are two major pathways of apoptosis activation, extrinsic (death receptor-dependent) and intrinsic (mitochondrial; coupled to cytochrome *c* release). The former pathway is activated by caspase-8/10 [31,32,33,34,35], while the latter requires caspase-9 [36,37,38,39,40]. Moreover, cleavage of poly(ADP-ribose) polymerase (the PARP protein) also stimulates the intrinsic apoptosis process [41,42]. Therefore, since the stimulation of apoptosis in cells affected by genetic defects, like Morquio disease, is assumed to proceed through the intrinsic pathway, and as we observed enhanced cytochrome *c* release from mitochondria in MPS IVA and MPS IVB cells (Figure 1), we investigated the levels of specifically cleaved caspases (caspase-9, caspase-3, caspase-6, and caspase-7) and PARP in Morquio and control cells, either untreated or treated with 1 μM staurosporine for 6 h. This time of treatment allowed for us to observe the middle time of the induced apoptosis. As shown in Figure 2, staurosporine efficiently induced apoptosis in both the control and Morquio disease cells, as monitored by the elevated levels of specifically cleaved caspases and PARP. However, while the levels of caspase-9 (cleaved at Asp330) were higher in control (HDFa) cells than in Morquio fibroblasts, those of caspase-9 (cleaved at Asp315) were similar in all the tested cell lines treated with staurosporine. Moreover, during the staurosporine-mediated induction of apoptosis, the levels of caspase-3 (cleaved at Asp175), caspase-6 (cleaved at Asp162), caspase-7 (cleaved at Asp198), and PAPR (cleaved at Asp214) were higher in MPS IVA and MPS IVB than in HDFa (Figure 2). Importantly, in non-induced cells, the levels of caspase-3 (cleaved at Asp175) and PAPR (cleaved at Asp214) were increased in MPA IVA (but not in MPS IVB) relative to the control cells (Figure 2). These results confirm that apoptosis is up-regulated in Morquio disease and suggest that the execution is the main stage of this process, which is more efficient in MPS IVA and IVB cells relative to the control cells.

## 3. Discussion

Apoptosis is the process of programmed cell death. It plays crucial roles in many biological processes, including the development of different organs and the prevention of carcinogenesis [28,29,30]. Therefore, both impairments and overstimulation of this process may be deleterious for cells, and thus for whole organisms.

Although MPSs are monogenic diseases, their mechanisms are complex and include not only the primary storage of GAGs but also secondary and tertiary effects, which interfere with various biochemical pathways and lead to significant changes in different cellular processes [13,14,15,16,17,18,19,20,21,22]. This makes understanding the molecular mechanisms of the disease considerably more difficult than previously estimated, when considering the accumulation of GAGs as the only cause of the symptoms. 

One of processes that was found to be dysregulated in MPS is apoptosis, which has been preliminarily demonstrated in transcriptomic studies to be stimulated relative to control cells [22]. One might assume that too effective programmed cell death should lead to unnecessary loss of cells, which might contribute to the pathomechanism of the disease. If such a hypothesis is true, it might partially explain the underdevelopment of some organs, due to the decreased number of cells required to build them. 

In Morquio disease, skeleton and connective tissue especially are severely affected [23], and thus, it is tempting to speculate that the enhanced efficiency of apoptosis could participate in the development of specific problems with bones, cartilages, and connective tissue occurring in this disorder. Intriguingly, although there were some reports in the literature on the changes in apoptosis in different MPS types (see below for details), there were only a few reports in which apoptosis was investigated in Morquio disease, while all of studies concerned various types rather than specifically MPS IV [22,33]. Therefore, in this work, we have focused on changes in apoptosis in fibroblasts derived from Morquio disease patients. Generally, there are two signaling pathways leading to induction of apoptosis [28,29,30]. One of them is mediated by external signals and death receptors, while the second is dependent on internal signals and the release of cytochrome *c*. The former pathway requires caspases-8/10 at the initiation stage, while the latter is dependent on activation of caspase-9 [31,32,33,34,35,36,37,38,39,40]. Since MPS IV is an example of a genetic disease, and any changes are expected to be mainly due to internal perturbations in cellular structures and/or processes, we focused on the latter pathway of apoptosis, and thus, on determining cytochrome *c* release and activation of caspase-9, and then on levels of executive caspases (3, 6, and 7).

Our results corroborated previously reported predictions [22] that apoptosis is stimulated in MPS IVA and IVB (Table 1, Figure 1). Importantly, such an elevated efficiency of apoptosis was found in both cells growing under standard laboratory conditions (as revealed by the cytochrome *c* release from mitochondria, while increased levels of cleaved caspase-3 and PARP were observed only in MPS IVA, not in MPS IVB) and after staurosporine-mediated induction (evidenced in both tests, cytochrome *c* release and elevation of levels of cleaved caspases and PARP) in MPS IVA as well as MPS IVB (Figure 1 and Figure 2). Moreover, it was possible to estimate that the MPS IV-specific stimulation occurs at the stage of the action of executive caspases and PARP, rather than during the initiation step (Figure 2). It might be intriguing as to why effects of the MPS IVA and IVB disease on the cleavage of caspase-9 at two different positions, Asp315 and Asp330, are different (Figure 2; a lack of differences between control cells and MPS IV fibroblasts in the case of the cleavage at Asp315 and significant differences in the case of that at Asp330). One possible explanation might be that, in the initial proteolytic reaction, the result of the auto-cleavage at Asp315, catalyzed by the pro-caspase-9 itself, activates the molecular timer of apoptosis. On the contrary, the cleavage at Asp330 is mediated by activated caspase-3, according to the positive feedback mechanism [28,29,30]. Therefore, these results might strengthen the proposal that MPS IV-specific changes in the apoptosis process, relative to normal cells, arise not at the initiation stage but at the further steps of the pathway. 

As mentioned above, some works indicating changes in apoptosis in different MPS types were previously published, with only preliminary information on MPS IV [22,43]. Nevertheless, since all MPS types are primarily caused by the accumulation of GAGs, it is worth comparing the previously reported results of studies on types other than Morquio disease. MPS I appears to be the most extensively studied type in this aspect, though nonetheless only a few articles were published. Using a murine model of MPS I, it was postulated that lysosomal membrane permeabilization might occur, which was accompanied by an increased rate of apoptotic cells relative to controls [44]. Subsequent studies, conducted using mouse cells, corroborated this observation while suggesting that the induction of apoptosis in this disease may arise from deficiency in α-L-iduronidase, which indirectly leads to down regulation of the *bcl-2* gene [45]. Employing MPS I murine granulocyte-macrophage progenitor cells, it was then observed that the affected cells revealed higher sensitivity to staurosporine treatment [46]. On the other hand, no significant changes in apoptosis could be detected in murine MPS I fibroblasts relative to controls [47]. Nevertheless, a very recent study with a zebrafish model of MPS I provided evidence for an increased efficiency of apoptosis in α-L-iduronidase-deficient larvae [48].

Changes in apoptosis in MPS II were mainly investigated with neural cell models, which demonstrated significant changes in this process relative to controls. Specifically, premature apoptosis of MPS II murine neural stem cells was observed [49], and increased efficiency in this process was reported in human neuronal cells as estimated by determining the levels of caspases and ultrastructural changes in cells [50].

When studying brains of the mouse model of MPS IIIA, changes in expression of genes encoding proteins related to apoptosis (like TNF-α, TNFR1, caspase-3, caspase-11) were reported [51]. Similarly, more apoptotic cells were found in brains of MPS IIIB mice than in brains of healthy mice [52]. 

In MPS VI, an elevated number of apoptotic chondrocytes derived from affected rats and cats, as compared to those isolated from healthy animals, was observed [53]. Interestingly, this phenomenon was proposed to be responsible for the joint disease in MPS VI, which is intriguing in light of the results reported in this work for Morquio disease, wherein joints are heavily affected. Similar observations in MPS VI rats were also published subsequently [54,55]. Moreover, an iPSC-based model of cartilage pathology in MPS VI allowed for the observation of the up-regulation of expression of genes involved in apoptosis [56]. Marked chondrocyte apoptosis was also observed in MPS VI and MPS VII [57]. Intriguingly, it appears that increased levels of dermatan sulfate, a GAG stored in MPS VI, are not sufficient for the stimulation of apoptosis in MPS VI [58].

The stimulation of apoptosis in MPS VII, mentioned in the preceding paragraph, was also demonstrated in other studies. Namely, the expression of genes coding for apoptotic proteins (TNF-R1, caspase-3, caspase-9, caspase-11) was enhanced in the brains of MPS VII mice [59], and apoptosis stimulation was evident also in the resting zone of the growth plate, contributing to the pathology of bones and cartilages [60]. Extensive apoptosis was also reported in the neurons of the *Drosophila* model of MPS VII [61].

In summary, the previously published results (and summarized above) indicate that apoptosis is up-regulated in different types of MPS, including types I, II, IIIA, IIIB, VI, and VII. Despite some preliminary results, derived from transcriptomic analyses of the material from cells representing several MPS types, suggesting that apoptosis might also be stimulated in MPS IVA and IVB [22], no studies have specifically addressed this problem in Morquio disease. Cytotoxicity assays and proteomic analyses even suggested that apoptosis might be unchanged in MPS IVA [43,62]. Here, we experimentally demonstrate that, similarly to other MPS types, enhanced efficiency of the basal and induced apoptosis process also occurs in MPS IVA and IVB fibroblasts. It is likely that the enhanced programmed cell death may contribute significantly to the pathomechanism of Morquio disease.

## 4. Materials and Methods

### 4.1. Cell Cultures

Fibroblasts derived from patients suffering from MPS IVA and IVB, and the control fibroblast line (HDFa) was purchased from the NIGMS Human Genetic Cell Repository at the Coriell Institute for Medical Research (Camden, NJ, USA). MPS cells bear the following mutations: MPS IVA, p.Arg386Cys/p.Phe285Ter (in the *GALNS* gene), and MPS IVB, p.Trp273Leu/p.Trp509Cys (in the *GLB1* gene). Cells were cultured in the DMEM medium (Dulbecco’s modified Eagle’s, Gibco Ltd., London, UK), supplemented with 10% (*v*/*v*) FBS (fetal bovine serum) and standard penicillin-streptomycin mixture, at 37 °C, 95% humidity, 5% CO_2_ saturation.

### 4.2. Transcriptomic Studies

Transcriptomic analyses were conducted using previously described methods [13]. Briefly, cells were cultured, lysed, and homogenized for RNA extraction. The quality of RNA was assessed, and libraries of mRNAs were prepared. Sequencing was performed using HiSeq4000 (Illumina, San Diego, CA, USA) (PE150; 150 bp paired-end) and yielded at least 4 × 10^7^ raw reads per sample. The data quality was checked with FastQC (version v0.11.7). Raw reads were mapped to the GRCh38 human reference genome using Hisat2 v.2.1.0 software. Transcript levels were calculated with Cuffquant and Cuffmerge software (version 2.2.1). The raw results are available under accession no. PRJNA562649 in the NCBI Sequence Read Archive (SRA). Statistical analyses utilized one-way ANOVA and post hoc Student’s t-test with Bonferroni correction. The false discovery rate (FDR) was determined using the Benjamini–Hochberg method. Transcripts were classified using the Ensembl gene database through the BioMart interface (https://www.ensembl.org/info/data/biomart/index.html; accessed on 5 December 2021).

### 4.3. Immunofluorescence Microscopy

Fibroblasts (4 × 10^4^ cells) were cultured overnight on coverslips. Subsequently, cells were incubated for 5.5 h in the presence of either 1 μM staurosporine or an equivalent concentration of DMSO as a control. Following such incubation, MitoTracker™ Orange CMTMRos (#M7510; Invitrogen, Waltham, MA, USA) was added according to the manufacturer’s instructions and incubation was prolonged for an additional 0.5 h. The cells were then fixed using 2% paraformaldehyde for 15 min at 37 °C, followed by washing with 0.1% Triton X-100 for 15 min at 37 °C. Next, the cells were blocked with a BSA solution for 1 h at room temperature, and subsequently incubated overnight with anti-cytochrome *c* antibody, diluted 1:600 (#12963; Cell Signaling Technology Inc., Danvers, MA, USA). Afterward, secondary antibody, conjugated with Alexa Fluor™ 488 (Thermo Fisher Scientific, Waltham, MA, USA), was used at a dilution of 1:1000 and samples were incubated for 1 h at room temperature. Finally, the coverslips were mounted on slides and the prepared microscope slides were analyzed using a confocal laser scanning microscope from Olympus (Tokyo, Japan). The release of cytochrome *c* from mitochondria was estimated by measurement of the green fluorescence signal which did not colocalize with the fluorescence (red pseudocolor) derived from MitoTracker™ Orange CMTMRos, as described previously [63]. The experiment was repeated 3 times, with 100 cells analyzed in each sample. Statistical analyses were performed using two-way ANOVA with Tukey’s post hoc test. Differences were considered significant when *p* < 0.05.

### 4.4. Western Blotting

Fibroblasts (5 × 10^5^ cells) were seeded on 100 mm dishes, where they were allowed to grow overnight. The following day, cells were treated with either 1 µM staurosporine or an equivalent concentration of DMSO (control) for 6 h, and then collected. After harvesting, the cells were lysed for 30 min on ice, using a lysis solution (50 mM Tris (pH 7.5), 1% Triton X-100, 150 mM NaCl, 0.5 mM EDTA) and protease and phosphatase inhibitor cocktails (Roche Diagnostics, Vienna, Austria). The lysates were centrifuged, and proteins were separated and detected using the WES system (Automated Western Blots with Simple Western; ProteinSimple, San Jose, CA, USA) for automatic Western blotting. For the protein separation, the 12–230 kDa Separation Module with 8 × 25 capillary cartridges (#SM-W004; ProteinSimple, San Jose, CA, USA) was utilized. The detection of specific proteins was achieved using antibodies purchased from Cell Signaling Technology, Inc. (Danvers, MA, USA), including those detecting cleaved caspase-3 Asp175 (#9664), cleaved caspase-6 Asp162 (#9761), cleaved caspase-7 Asp198 (#8438), cleaved caspase-9 Asp330 (#52873), cleaved caspase-9 Asp315 (#9505), and cleaved PARP Asp 214 (#5625); all these antibodies were diluted 1:50. For signal detection, secondary goat anti-rabbit antibody (#7074, Cell Signaling Technology) and the Anti-Rabbit Detection Module (#DM-001, ProteinSimple) were used. The Total Protein Detection Module for Chemiluminescence (#DM-TP01, ProteinSimple) was employed to detect total protein levels in samples, assessed as a loading control. Protein levels were assessed using the WES system, and ImageJ software included into the WES system was utilized for the analysis of protein levels. Each experiment was repeated 3 times, with protein samples obtained from 3 independent cultures analyzed in each repetition. The data were analyzed using two-way ANOVA with Tukey’s post hoc test. Differences were considered significant when *p* < 0.05

## 5. Conclusions

The apoptosis process was stimulated in MPS IVA and IVB fibroblasts relative to control cells, as suggested previously on the basis of preliminary transcriptomic analyses [22] and demonstrated experimentally for the first time in this report. This stimulation occurred under both standards laboratory conditions (basal level of apoptosis) and after induction with staurosporine. The enhancement of apoptosis in Morquio disease occurs most probably at the stage of executive actions of caspases and PARP, of which levels are increased. This phenomenon may contribute to the pathomechanism of the disease, especially when bones, cartilage and connective tissue are formed.

## Figures and Tables

**Figure 1 ijms-24-14119-f001:**
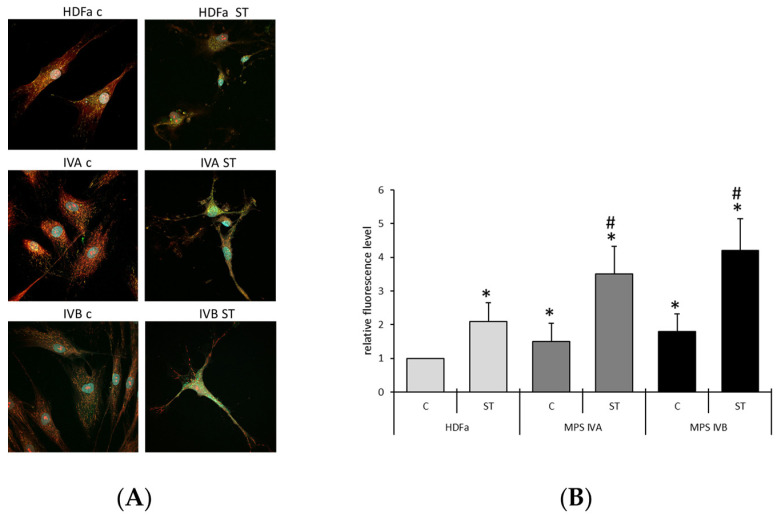
Cytochrome *c* release from mitochondria before and during staurosporine-induced apoptosis. Cells (HDFa, MPS IVA or MPS IVB) were cultured in the absence (c) or presence (ST) of 1 μM staurosporine for 6 h. Fibroblasts were stained with DAPI (blue), MitoTracker Orange (pseudocolored red), and Cytochrome Mouse mAb with fluorochrome-conjugated anti-mouse secondary antibody (green). Representative micrographs are shown (**A**), and quantification of the released cytochrome *c* was performed by measuring of green fluorescence intensity; scale bar: 10 mM (**B**). Experiments were repeated 3 times with 100 randomly chosen cells analyzed in each sample. Statistically significant differences (*p* < 0.05 in two-way ANOVA with Tukey’s post hoc) relative to HDFa-c (*) or between MPS IVA-ST or MPS IVB-ST and HDFa-ST samples (#) are shown.

**Figure 2 ijms-24-14119-f002:**
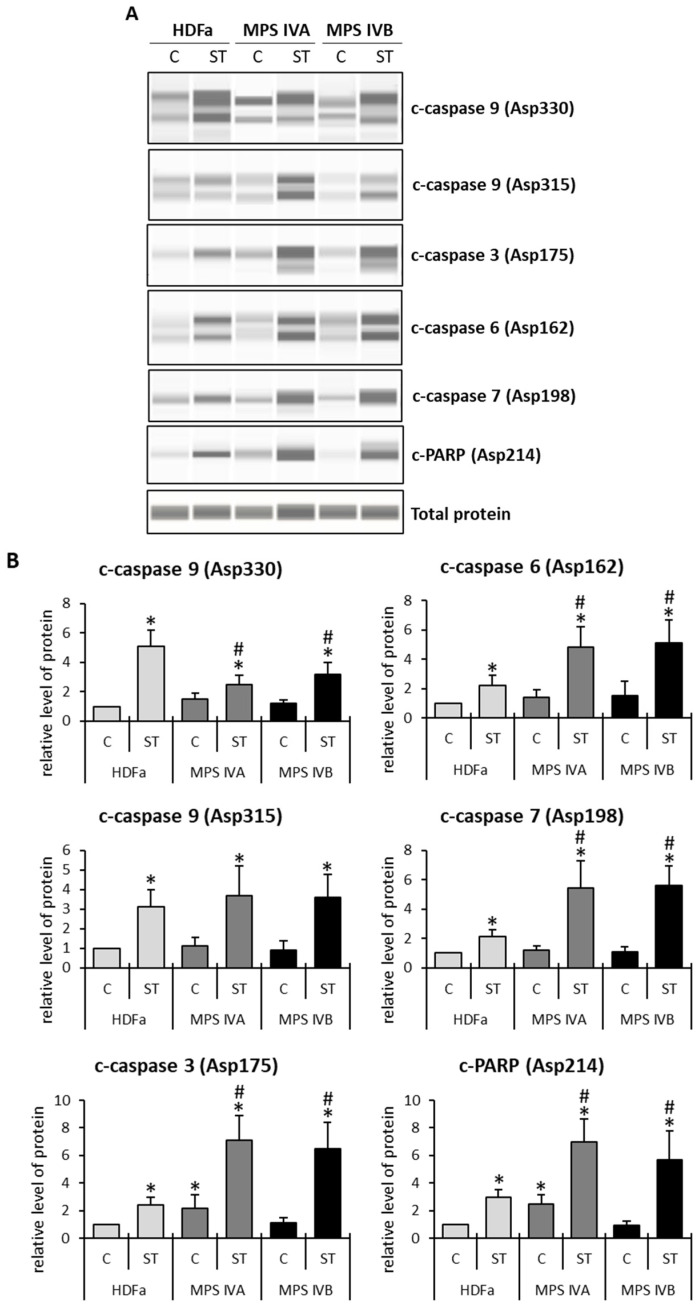
Relative levels of cleaved proteins in cells with staurosporine-induced apoptosis. Cells (HDFa, MPS IVA or MPS IVB) were cultured in the absence (C) or presence (ST) of 1 μM staurosporine 6 h. Protein levels were assessed using the automatic Western blotting (the WES system, with capillary electrophoresis). In panel (**A**), representative blots of cleaved, apoptosis-related proteins are shown. In panel (**B**), each column presents the mean value (with standard deviation indicated) derived from three independent experiments, after densitometric quantification of the intensities of bands in the blots. Statistically significant differences (*p* < 0.05 statistically significant differences (*p* < 0.05 in two-way ANOVA with Tukey’s post hoc) relative to HDFa-C (asterisks) or between MPS IVA-ST or MPS IVB-ST and HDFa-ST samples (hashtags) are indicated.

**Table 1 ijms-24-14119-t001:** Genes included in the term “apoptotic process” in Gene Ontology (http://geneontology.org/; last assessed on 27 July 2023) (GO:0006915) with expression significantly changed (at FDR < 0.1; *p* < 0.1) in MPS IVA and IVB fibroblasts relative to control cells (HDFa). Transcriptomic studies were performed using RNA-seq analysis, and log_2_ of fold change (FC) values are shown.

Log_2_FC of Levels of Selected Transcripts in MPS IV Subtypes vs. HDFa Line					
Transcript *	MPS IVA	MPS IVB	Transcript *	MPS IVA	MPS IVB
Up-Regulated Transcripts			Down-Regulated Transcripts		
*CLU* (tr. 1)	**2.26**	**3.84**	*BCAP29*	**−1.02**	**−0.90**
*CLU* (tr. 2)	1.73	**3.46**	*BNIP3*	**−0.71**	**−0.74**
*CLU* (tr. 3)	1.11	**2.89**	*C1D*	**−1.38**	**−1.38**
*CLU* (tr. 4)	1.90	**3.43**	*CDKN1A*	**−2.42**	−2.40
*CLU* (tr. 5)	1.86	**3.46**	*CLPTM1L*	**−1.33**	−1.04
*CRYAB*	**0.38**	−0.31	*GAPDH*	**−0.67**	−0.78
*COMP*	2.58	**1.67**	*GPER1*	**−0.96**	**−1.51**
*CRIP1*	3.18	**2.01**	*IGFBP3*	**−5.17**	−1.49
*ERCC2*	0.47	**0.39**	*PRKCD*	**−0.65**	**−0.90**
*ERCC6*	0.72	**0.97**	*RYBP*	**−0.77**	**−0.76**
*FNIP2*	0.92	**0.97**	*SGK1*	**−1.33**	0.18
*HIC1*	0.35	**0.74**	*TNFAIP8*	**−0.52**	−0.26
*HIP1*	0.84	**0.99**	*ACAA2*	−0.20	**−0.59**
*MAX*	1.11	**1.33**	*ARL6IP5*	−0.38	**−0.72**
*MLLT11*	0.78	**0.69**	*CAV1* (tr. 1)	−0.31	**−0.89**
*MUL1*	0.10	**0.64**	*CAV1* (tr. 2)	−0.40	**−1.05**
*NLRP1*	0.84	**1.85**	*CAV1* (tr. 3)	−2.95	**−4.74**
*PLK3*	0.50	**0.59**	*CYCS*	−0.81	**−0.76**
*PPARD*	0.38	**0.57**	*FOXO3*	−0.57	**−0.51**
*RHOB*	0.52	**0.94**	*GAS6*	−0.39	**−1.29**
*RNF216*	0.38	**0.44**	*HTATIP2*	0.02	**−0.97**
*RPS3*	0.97	**1.00**	*KPNB1*	−1.99	**−1.58**
*SMAD3*	0.61	**1.21**	*KREMEN1*	−1.77	**−2.54**
*STPG1*	−0.19	**0.72**	*MSH5*	−0.93	**−1.33**
*TMEM109*	0.29	**0.33**	*PDK1*	−0.90	**−1.39**
*TOP2A*	1.64	**1.68**	*PDK2*	−0.73	**−0.80**
*TRIM35*	0.29	**0.51**	*PERP*	−0.55	**−0.78**
*UACA*	0.21	**0.99**	*PLSCR1*	−0.44	**−0.67**
*XIAP*	0.48	**1.15**	*PSMD10*	−0.37	**−0.45**
*ZC3H12A*	0.71	**1.02**	*RIPK2*	0.11	**−0.56**
			*SGPP1*	−0.42	**−0.22**
			*SH3GLB1* (tr. 1)	−0.15	**−0.31**
			*SH3GLB1* (tr. 2)	−0.63	**−0.63**
			*SOD2*	−1.34	**−1.73**
			*TGFBR1*	−0.26	**−1.09**
			*TGFBR2*	−0.33	**−0.74**
			*TMBIM4*	−0.46	**−0.92**
			*TNFRSF11B*	−0.74	**−1.23**
			*TRAF3IP2*	−0.33	**−1.04**

* For some genes, more than one transcript could be detected; in such cases, separate transcripts are marked as tr.1, tr.2, etc. Transcripts in which levels were significantly different (*p* < 0.1) in MPS IVA or IVB fibroblasts relative to those measured in control cells (HDFa) are marked in bold.

## Data Availability

Transcriptomic (RNA-seq) raw results are available in the NCBI Sequence Read Archive (SRA), under accession no. PRJNA562649mRNA. Other raw results are available from the authors upon request.

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
