# Peer review of "Enhanced Efficiency of the Basal and Induced Apoptosis Process in Mucopolysaccharidosis IVA and IVB Human Fibroblasts"

_ijms, 2023, doi:10.3390/ijms241814119_

Round 1

Reviewer 1 Report

Mucopolysaccharidosis is one of lysosomal storage diseases. It is known that patients with Morquio disease (MPSIV) have no or little cognitive impairment and severe skeletal deformities. Now, there's an enzyme replacement therapy like elosulfase alfa for MPS IV. But the effectiveness of the ERT is limited. So much further elucidation of pathology and development of new therapies are to be expected. As the author mentioned, MPS IV is the leaset studied MPS type in the light of apoptosis dysregulation. So it is a good point for the author to try to show dysregulation of apoptosis in MPS IV.

Comments

#1 As for Figure 1, the author shows the relative fluorescence levels compared to HDFa. However, the images of HDFa ST, IVA ST and IVB ST differ from one another, although those of IVA c and IVBc are exception. I don't think they are compareble at a glance.

Author Response

COMMENTS OF THE REVIEWER:

Mucopolysaccharidosis is one of lysosomal storage diseases. It is known that patients with Morquio disease (MPSIV) have no or little cognitive impairment and severe skeletal deformities. Now, there's an enzyme replacement therapy like elosulfase alfa for MPS IV. But the effectiveness of the ERT is limited. So much further elucidation of pathology and development of new therapies are to be expected. As the author mentioned, MPS IV is the leaset studied MPS type in the light of apoptosis dysregulation. So it is a good point for the author to try to show dysregulation of apoptosis in MPS IV.

Comments

#1 As for Figure 1, the author shows the relative fluorescence levels compared to HDFa. However, the images of HDFa ST, IVA ST and IVB ST differ from one another, although those of IVA c and IVBc are exception. I don't think they are compareble at a glance.

RESPONSE:

Indeed, as noted by the reviewer, there are differences between cells shown in particular images. These differences might arise from both the Morquio disease which influences morophology of organelles [Gaffke et al. Int J Mol Sci 2021, 22, 2766], thus, these cells differ from control ones, and the effects of staurosporine. Please, note that the response to staurosporine was also different between various tested cell lines. Moreover, it is also worth noting that the Figure shown representative cells, while the analyses were performed with 100 randomly selected cells in each experiment (as indicated in Materials and Methods). Finally, it is crucial to note that fluorescence signals were parameters which were measured and analyzed, thus, even if cell morphology was different, the signals (indicating release of cytochrome c from mitochondria) could be comparted between different cell lines.

Reviewer 2 Report

Brokowska and colleagues present that cytochrome c release from mitochondria is more efficient in MPS IVA and IVB fibroblasts compared to control cells, both under standard cultivation conditions and after treatment with staurosporine, an apoptosis inducer. However, this manuscript requires significant editing before publication (see details below).

Introduction

1. Page 1, Line 41: "They include types I, II, IIIA, IIIB, IIIC, IIID, IIIE (until recently known only in mouse models, but patients with mutations in the ARSG gene, coding for arylsulfatase G which is defective in this type, have recently been described, though sometimes diagnosed for different diseases [5,6]), IVA, IVB, VI, VII, IX, X." The mutation of ARSG gene in humans presents as Usher syndrome type IV. MPS IIIE is now noted as a formal subtype in humans.

Results

1. Page 4, Line 128: "Therefore, we investigated levels of specifically cleaved caspases (caspase 9, caspase 3, caspase 6, and caspase 7) and PARP in Morquio and control cells, either untreated or treated with 1uM staurosporine for 6h." I am curious why caspase-8 was not chosen for the experiment.

2. Page 4, Line 133: "However, while levels of caspase-9 (cleaved at Asp330) were higher in control (HDFa) cells than in Morquio fibroblasts, those of caspase-9 (cleaved at Asp315) were similar in all tested cell lines treated with staurosporine." I am curious why the same caspase with different Asp cleavages showed different results.

Conclusion

1. Page 9, Line 324: "The apoptosis process is stimulated in MPS IVA and IVB fibroblasts relative to control cells, as suggested previously on the basis of preliminary transcriptomic analyses [22] and demonstrated experimentally for the first time in this report." [22) should be [22].

Minor edits to the English language are required.

Author Response

REVIEWER’S COMMENT

Introduction

  1. Page 1, Line 41: "They include types I, II, IIIA, IIIB, IIIC, IIID, IIIE (until recently known only in mouse models, but patients with mutations in the ARSG gene, coding for arylsulfatase G which is defective in this type, have recently been described, though sometimes diagnosed for different diseases [5,6]), IVA, IVB, VI, VII, IX, X." The mutation of ARSG gene in humans presents as Usher syndrome type IV. MPS IIIE is now noted as a formal subtype in humans.

RESPONSE:

We thank the reviewer for this comment. Indeed, MPS IIIE has recently been formally recognized as a human disease. Therefore, we have revised the sentence which now reads as follows (lines 40-45):

“They include types I, II, IIIA, IIIB, IIIC, IIID, IIIE (recognized recently as a deficiency in  arylsulfatase G, encoded by the ARSG gene [5,6]), IVA, IVB, VI, VII, IX, X (the latest type described [7]), and MPSPS (MPS-Plus syndrome, caused by mutations in the VPS33A gene rather than in a gene coding for a lysosomal hydrolase, though resulting in massive GAG accumulation [8,9]).”

REVIEWER’S COMMENT

Results

  1. Page 4, Line 128: "Therefore, we investigated levels of specifically cleaved caspases (caspase 9, caspase 3, caspase 6, and caspase 7) and PARP in Morquio and control cells, either untreated or treated with 1uM staurosporine for 6h." I am curious why caspase-8 was not chosen for the experiment.

RESPONSE:

We thank the reviewer for this important question. In fact, in the original paper, we did not present a sufficient information on the rationale of choosing caspases which were tested. Now, such an explanation is provided (lines: 188-195) and reads as follows:

“Generally, there are two signaling pathways leading to induction of apoptosis [28-30]. One of them is mediated by external signals and death receptors, while the second is depend-ent on internal signals and the release of cytochrome c. The former pathway requires caspases-8/10 at the initiation stage, while the latter is dependent on activation of caspase-9 [28-30]. Since in MPS IV, as an example of a genetic disease, any changes are expected to be mainly due to internal perturbations of cellular structures and/or processes, we focused on the latter pathway of apoptosis, and thus, on determining the cytochrome c release and activation of caspase-9, and then on levels of executive caspases (3, 6, and 7).”

REVIEWER’S COMMENT

  1. Page 4, Line 133: "However, while levels of caspase-9 (cleaved at Asp330) were higher in control (HDFa) cells than in Morquio fibroblasts, those of caspase-9 (cleaved at Asp315) were similar in all tested cell lines treated with staurosporine." I am curious why the same caspase with different Asp cleavages showed different results.

RESPONSE:

This was another very interesting question. We have addressed this issue, and discussed the results in the revised manuscript as follows (lines 205-214):

“It might be intriguing why effects of the MPS IVA and IVB disease on the cleavage of caspase-9 at two different positions, Asp315 and Asp330 are different (Figure 2; a lack of differences between control cells and MPS IV fibroblasts in the case of the cleavage at Asp315 and significant differences in the case of that at Asp330). One possible explana-tion might be that the initial proteolytic reaction the result of the auto-cleavage at Asp315, catalyzed by the pro-caspase-9 itself, which activates the molecular timer of apoptosis. On the contrary, the cleavage at Asp330 is mediated by activated caspase-3, according to the positive feedback mechanism [28-30]. Therefore, these results might strengthen the pro-posal that MPS IV-specific changes in the apoptosis process, relative to normal cells, arise not at the initiation stage but at the further steps of the pathway.”

REVIEWER’S COMMENT

Conclusion

  1. Page 9, Line 324: "The apoptosis process is stimulated in MPS IVA and IVB fibroblasts relative to control cells, as suggested previously on the basis of preliminary transcriptomic analyses [22] and demonstrated experimentally for the first time in this report." [22) should be [22].

RESPONSE:

The text has been corrected as indicated by the reviewer (line 338 in the revised manuscript).

Round 2

Reviewer 2 Report

Thank you for sharing the author's responses to the reviewer's comments. Here are some additional suggestions that could further improve the manuscript:

1. In the Results section where caspase-8 is discussed, the authors could cite a couple of relevant references on the intrinsic and extrinsic apoptosis pathways to support their rationale.

Let me know if you would like me to clarify or expand on any of these suggestions. I'm happy to provide additional feedback to help strengthen this manuscript further.

Minor editing of English language required

Author Response

REVIEWER’S COMMENT

In the Results section where caspase-8 is discussed, the authors could cite a couple of relevant references on the intrinsic and extrinsic apoptosis pathways to support their rationale.

RESPONSE

As requested by the reviewer, we have included references relevant to intrinsic and extrinsic apoptosis pathways in the Results section. This strengthen the rationale presented in the paper, indeed. We thank the reviewer for this suggestion. The new text reads as follows (lines: 123-133):

“There are two major pathways of apoptosis activation, extrinsic (death receptor-dependent) and intrinsic (mitochondrial; coupled to cytochrome c release). The former pathway is activated by caspase-8/10 [31-35] while the latter requires caspase-9 [36-40]. Moreover, cleavage of poly(ADP-ribose) polymerase (the PARP protein) also stimulates the intrinsic apoptosis process [41,42]. Therefore, since stimulation of apoptosis in cells affected by genetic defects, like Morquio disease, is assumed to proceed through the intrinsic pathway and we observed enhanced cytochrome c release from mitochondria in MPS IVA and MPS IVB cells (Figure 1), we investigated levels of specifically cleaved caspases (caspase-9, caspase-3, caspase-6, and caspase-7) and PARP in Morquio and control cells, either untreated or treated with 1 mM staurosporine for 6 h.”